# FAK Structure and Regulation by Membrane Interactions and Force in Focal Adhesions

**DOI:** 10.3390/biom10020179

**Published:** 2020-01-24

**Authors:** Paula Tapial Martínez, Pilar López Navajas, Daniel Lietha

**Affiliations:** Centro de Investigaciones Biológicas (CIB), Spanish National Research Council (CSIC), Cell Signalling and Adhesion Group, 28040 Madrid, Spain; paula.tapial@cib.csic.es (P.T.M.); pilar.lopez.navajas@cib.csic.es (P.L.N.)

**Keywords:** focal adhesion kinase, cell adhesion, cell signalling, mechanobiology, membrane interactions, structural biology

## Abstract

Focal adhesion kinase (FAK) is a non-receptor tyrosine kinase with key roles in the regulation of cell adhesion migration, proliferation and survival. In cancer FAK is a major driver of invasion and metastasis and its upregulation is associated with poor patient prognosis. FAK is autoinhibited in the cytosol, but activated upon localisation into a protein complex, known as focal adhesion complex. This complex forms upon cell adhesion to the extracellular matrix (ECM) at the cytoplasmic side of the plasma membrane at sites of ECM attachment. FAK is anchored to the complex via multiple sites, including direct interactions with specific membrane lipids and connector proteins that attach focal adhesions to the actin cytoskeleton. In migrating cells, the contraction of actomyosin stress fibres attached to the focal adhesion complex apply a force to the complex, which is likely transmitted to the FAK protein, causing stretching of the FAK molecule. In this review we discuss the current knowledge of the FAK structure and how specific structural features are involved in the regulation of FAK signalling. We focus on two major regulatory mechanisms known to contribute to FAK activation, namely interactions with membrane lipids and stretching forces applied to FAK, and discuss how they might induce structural changes that facilitate FAK activation.

## 1. Introduction

Focal adhesion kinase (FAK) was discovered almost three decades ago as a kinase that is highly phosphorylated in response to cell adhesion and localisation into the focal adhesion complex [1,2]. Its importance in physiology and disease was quickly recognised and a host of cellular and in vivo studies have since placed FAK as a key player of cellular processes which in some form usually require controlled cell motility. As such, FAK is required for various aspects during development [3,4,5,6] as well as tissue regeneration and wound healing [7,8,9,10]. The importance of FAK in cell motility is also reflected in disease where FAK plays major roles in tumour invasion and metastasis [11]. In mouse models, FAK is shown to be responsible for skin and breast cancer invasion as well as breast to lung metastasis [12,13]. In patients high FAK expression is associated with poor prognosis and is shown to be a cause for resistance to a number of primary treatments [14,15,16,17,18].

FAK is today recognised as a hub in the interactome of focal adhesions [19,20,21], a protein complex at the centre of mesenchymal cell migration. Focal adhesions form on the cytoplasmic side of the plasma membrane at sites where integrin receptors cluster as they attach outside the cell to the extracellular matrix (ECM). Inside the cell, the focal adhesion complex is via actomyosin stress fibres connected to the actin cytoskeleton, and this structural ECM-actin link provides a means for the cell to gain traction during cell migration as actomyosin stress fibres contract. For the integrins to bind ECM they need to adopt an open conformation that exhibits high affinity for ECM ligands. This can occur either by outside-in signalling whereby integrin binding to the ECM stabilises an open and active integrin conformation. On the other hand, a high-affinity integrin state initiating ECM binding can be triggered by the binding of intracellular components, such as talin, kindlin, and RIAM [22,23].

Super-resolution optical microscopy has defined a layered focal adhesion architecture with a signalling layer closest to the plasma membrane, a central force transduction layer and an actin regulatory layer, the site of actin attachment to the focal adhesion complex [24]. In addition to the structural role of focal adhesions in transmitting force for cell motility, the application of force to the complex also triggers important signalling cascades in the membrane proximal signalling layer. FAK is a key component of the focal adhesion signalling layer and is shown to directly interact with membrane lipids [25,26]. Force generation in focal adhesions is shown to activate FAK [27,28,29] and its signals regulate the controlled maturation and turnover of the focal adhesion complex, cell spreading and migration, and via extensive cross-talk also feed into proliferative and survival signals [11]. Although first identified as a focal adhesion protein, FAK is today also known to play important roles in the nucleus, endosomes, and adherens junctions (reviewed in [30]).

FAK contains an N-terminal FERM (band 4.1, erzin, radixin, moesin homology) domain a central kinase domain followed by a 220-residue proline-rich low-complexity region and a C-terminal focal adhesion targeting (FAT) domain (Figure 1a). Individual globular domains are structurally characterized [31,32,33,34] and a crystal structure of the FERM and kinase region reveals the mode of FAK autoinhibition [35]. In this review, we discuss the current structural understanding of the different FAK modules and how localization into the focal adhesion complex induce structural changes that result in switching FAK from its inhibited to an active state. We focus on conformational changes induced by membrane binding as well as anchoring to the actin cytoskeleton and discuss how these two anchor points might be involved in force activation of FAK.

## 2. FAK Structure

### 2.1. The FERM Domain

The FERM domain (residues 33–361) of FAK adopts a three lobed structure (lobes F1, F2, F3), with the three lobes arranged in a clover-leaf shaped assembly [31] (Figure 1b). The F1 lobe (residues 33–127) resembles a ubiquitin-like fold, the F2 lobe (128–253) is all helical and similar to acyl-CoA-binding protein and the F3 lobe (254–361) adopts a fold very similar to pleckstrin homology and the phosphotyrosine binding (PTB) domains. The clover-leaf assembly of the FAK FERM domain is overall very similar to the one seen in the ERM family members ezrin [36], radixin [37], and moesin [38], but different to the FERM domain of talin, which surprisingly is found to arrange its lobes in a linear arrangement [39]. An interesting peculiarity in the FAK FERM domain is that an extra 10 residues at the C-terminus of the FAK FERM domain form an integral part of the domain. This C-terminal extension folds back and interacts with the FERM F3 lobe at a site where other FERM F3 lobes as well as PTB domains interact with protein peptide ligands. The F3 lobes of the radixin and talin FERM domains bind at the corresponding site to cytoplasmic tails of ICAM2 and β-integrins [37,40]. This raises the question whether for FAK the site could be regulated, being occupied by the C-terminal FERM sequence in a ground state, but replaced to anchor FAK to cytoplasmic receptor tails upon recruitment and activation in focal adhesions.

Crystallised FERM domains contain most of the linker residues following the FERM domain (residues 362–399 or 362–405, [31]). This linker region is seen to be involved in two interactions with the FERM domain (Figure 1b). Linker residues immediately following the FERM domain (368–375), including a PxxP motif known to constitute a binding site for the Src SH3 domain, are seen in two crystal structures to interact with the FERM F3 lobe and in one FERM molecule with the cleft between the F1 and F3 lobe [31,41]. Further, several crystal structures reveal linker residues 394–401 to form a β-sheet interaction with the FERM F1 lobe. Notably, this includes the Y397 autophosphorylation site in the linker; the regulatory implication of this interaction is discussed in Section 2.4. Lastly, the F2 lobe contains a basic patch on its surface, consisting of a highly conserved KAKTLRK sequence which is shown to be important for FAK activity in cells [42]. This sequence is implicated in a number of interactions, including in binding to phosphorylated tails of the Met receptor [43], phosphoinositide lipids [25,26] (see Section 3), and via an intramolecular interaction to the C-terminal FAT domain [44] (see Section 2.4).

### 2.2. The Kinase Domain

The FAK kinase domain adopts a typical two lobed fold seen in all protein kinases with the ATP binding site sandwiched between the two lobes (Figure 1c). Within the C-terminal (C) lobe of the kinase, the activation loop extends over 21 residues (564–585), starting with the catalytic DFG motif that is involved in ATP binding. In the inactive state this loop is unphosphorylated and highly flexible, therefore in most FAK kinase structures the loop is not fully defined. Upon activation, Src family kinases phosphorylate residues Y576 and Y577 in the activation loop. The phosphorylated Y577 residue interacts with several conserved basic residues in the kinase C-lobe resulting in structuring the activation loop into a typical β-hairpin structure as seen in several other phosphorylated kinase structures [35]. The phosphorylated Y576 points away from the kinase and is responsible for preventing inhibition by the FERM domain (see Section 2.2). The FAK kinase is not observed to undergo a α-C-helix movement, known in several other kinase families to be involved in kinase regulation. Another important regulatory feature in other kinases is an approximately 180° rotation of the Asp and Phe residues in the DFG motif into a so-called “DFG out” conformation. Although a crystal structure of FAK in the “DFG-out” conformation exists bound to a type II inhibitor [45], it is not clear whether FAK adopts this conformation merely as a result of specific ligand interactions, or whether it might be part of a native regulation or catalytic cycle as proposed for other kinases including the Abl kinase [46].

### 2.3. C-Terminal Regions

The region C-terminal to the kinase (residues 686–917) is proline rich and predicted to be unstructured. It contains two PxxP motives, site I (P^712^PKP) interacts with the SH3 domain of p130Cas [47] and site II (P^874^PKKPP) with the SH3 domains of the small GTPase activating proteins GRAF and ASAP [48,49]. The extreme C-terminus forms a four-helix bundle comprising the FAT domain (residues 922–1050) (Figure 1d). The FAT domain interacts with other focal adhesion proteins and is thereby, as its name suggests, responsible for targeting FAK into the focal adhesion complex. The FAT domain is reported to interact with paxillin [50] and talin [51,52], but only for the paxillin interaction there is structural information available that shows that the FAT domain interacts via two faces with helical LD2 and LD4 motifs of paxillin [53,54]. One LD binding motif is formed by helices 1 and 4 (helix 1-4 face; with helix 1 being the most N-terminal and helix 4 the most C-terminal helix), the other by helices 2 and 3 (helix 2-3 face). A proline rich sequence immediately prior to the FAT domain (910–921), that includes the ERK phosphorylation site S910 [55], is seen in several structures to interact with the helix 1-4 face of the FAT domain, thereby partially overlapping with one of the paxillin binding sites. It is not clear whether this interaction occurs natively and since the overlap is rather small paxillin binding would likely displace residues 910–913 of the N-terminal FAT extension, possibly promoting ERK mediated S910 phosphorylation. The C-terminal region further contains two main tyrosine phosphorylation sites Y861 and Y925, both of which are phosphorylated by the Src kinase. Interestingly, Y925 is at the beginning of the first helix of the FAT domain, suggesting that the helix has to unfold to allow Y925 phosphorylation. To this end, in one crystal structure the first helix in FAT is swapped between two neighbouring FAT molecules in the crystal [33]. Subsequent studies have provided evidence for the propensity for this helix to detach from the four-helix bundle [56,57] and promote C-terminal phosphorylations on Y861, Y925 and S910 [58,59]. Mutations that promote destabilisation of the N-terminal FAT helix have further suggested helix-swap as a potential mode of FAT dimerization.

### 2.4. Regulatory Features in the FAK Structure

An important autoregulatory mechanism occurs through an interaction between the FERM and kinase domains to maintain an autoinhibited state (Figure 1e). As revealed in a crystal structure, a main interaction is formed between the FERM F2 lobe and the kinase C-lobe. The core of the interaction is formed in a key-in-look fashion, with F596 in the kinase domain inserting into a hydrophobic pocket on the FERM F2 lobe [35]. The mutation of either F596 or residues in the pocket on the FERM F2 lobe (e.g., Y180, M183) result in activated forms of FAK with increased catalytic activity and exposed regulatory phosphorylation sites Y397, Y576, and Y577. Using small angle scattering (SAXS) it was shown that a FERM-kinase fragment of FAK with the Y180A and M183A mutations adopts an open and elongated shape compared to the compact WT form [26]. In addition to the FERM-F2/kinase-C lobe interaction an indirect contact is formed in autoinhibited FAK between the FERM-F1 lobe and the kinase N-lobe, with a stretch of the linker bridging the two lobes [35]. This part of the linker includes the Y397 autophosphorylation site, which sandwiched between the two domains is inefficiently phosphorylated. The linker interactions with the FERM F1-lobe are formed by a β-sheet extension, identical as seen in FERM-linker structures [31] (compare Figure 1b to Figure 1e). It is noteworthy that once Y397 autophosphorylation occurs the FAK conformation or activity are not significantly affected [26,35]. However, Y397 autophosphorylation does convert the site into a high affinity binding site for the Src SH2 domain. Hence autophosphorylation is responsible for recruiting the Src kinase to FAK. Src interacts via its SH2 domain to pY397 in FAK and with the SH3 domain to the PxxP motif in the linker between the FERM and kinase domains of FAK. These interactions can contribute to Src activation and have been reported to recruit Src into focal adhesions [60]. Src, in turn phosphorylates several tyrosines in FAK, including Y576, Y577 in the activation loop of the FAK kinase, which induces full catalytical activity of FAK (see Section 2.2). They do so on the one hand by stabilizing an active conformation of the activation loop (pY577) which allows productive binding of ATP and Tyr-peptide substrates. On the other hand, the superposition of the phosphorylated kinase structure with autoinhibited FAK shows that the pY576 residue in the active kinase is not compatible with autoinhibitory FERM interactions and as a result Y576/Y577 phosphorylated FAK is fully active and no longer susceptible to FERM inhibition [35]. The effects of Y397 and Y576/Y577 phosphorylations on the FAK conformation were also shown using a purified form of a conformational FRET sensor with cyan fluorescent protein added N-terminally to the FERM domain and citrine inserted N-terminally to the kinase domain. These experiments show that there is little FRET change upon Y397 autophosphorylation, while Src phosphorylation of Y576/Y577 induces low FRET signals indicative of an open conformation [26].

Further, an intramolecular interaction has been demonstrated between the FAK FERM domain and the C-terminal FAT domain [44]. Using SAXS, mutagenesis and interaction studies it was shown that the FAT domain interacts with the KAKTLRK basic patch in the FERM F2 lobe. Surprisingly, this interaction appears to synergise with the FAT-paxillin interaction. The same authors demonstrate that FAK can dimerise via FERM F3 interactions. This involves the highly conserved W266 residue in the F3 lobe, which in all crystal structures containing the FERM domain is involved in a two-fold symmetric crystal contact equivalent to the one responsible for FAK dimerization (Figure 1f). Interestingly, the FAT/FERM interaction stabilises the FAK dimer. Since paxillin binding enhances the FAT/FERM interaction it is plausible that recruitment of FAK into focal adhesions triggers FAK dimerization, firstly by increasing the local FAK concentration, but also by promoting the FAT-FERM interaction. Although this study showed that W266 mediated dimerization increases trans-FAK autophosphorylation, it appears unlikely that this alone will trigger full FAK activation. The SAXS data showed that in the dimer FAK maintains the autoinhibitory FERM-kinase interaction indicating that auto- and Src-phosphorylations are still sub-optimal.

Likely, further external stimuli are needed for FAK activation. From a host of cell biology studies, it is long known that FAK is activated via both growth factor signalling and upon integrin mediated cell adhesion to the ECM. Several studies have suggested direct interactions between FAK and cytoplasmic portions of growth factor receptors, including Met, RET, EGFR and VEGEFR [43,61,62,63]. Interactions with phosphorylated tails of the Met receptor have been suggested to occur via the basic KAKTLRK region in the FERM domain. Yet, how growth factor receptors induce FAK activation is mechanistically still not understood. In oncogenic settings, and particular upon application of FAK inhibitors, it was shown that growth factor receptors can directly phosphorylate Y397 in FAK, thereby sustaining FAK scaffolding functions via phospho-Y397 even if FAK is efficiently inhibited [64]. Integrin mediated FAK activation is shown to be promoted by the plasma membrane lipid phosphatidylinositol 4,5-bisphosphate (PIP2) [26]. FAK recruited to focal adhesions via FAT-paxillin interactions binds PIP2 in the nearby plasma membrane, also via the KAKTLRK basic patch in the FERM F2 lobe. This interaction is possibly enhanced by the avidity of FAK dimers formed in focal adhesions. Membrane binding would likely replace the FERM-FAT interaction since both bind to the KAKTLRK basic patch. Binding of FAK to PIP2 membranes is then shown to promote formation of larger FAK oligomers as well as conformational changes that appear to expose the Y397 site for efficient autophosphorylation [26] (see Section 3). Membrane induced conformational changes were detected using the purified conformational FRET sensor described above. Possibly, membrane binding was also the primary cause for a FRET change in a cellular study that used a similar FRET sensor but found that the sensor did not require FAK activity or autophosphorylation for an optimal FRET response upon localisation into focal adhesions [65]. Introducing mutations at the KAKTLRK PIP2 binding site did indeed render the sensor insensitive. Alternatively, other interactions in focal adhesions might be responsible for FRET changes in this study.

## 3. FAK Association with the Membrane

### 3.1. FAK Integration into the Focal Adhesion Complex

Super-resolution optical microscopy has identified three distinct layers in the ultrastructural architecture of focal adhesions: a membrane proximal integrin signalling layer, an intermediate force transduction layer and a membrane distal actin regulatory layer [24]. FAK, localises into the signalling layer at the membrane. The membrane at focal adhesion sites contains increased levels of PIP2 due to the enzyme phosphatidylinositol-4-phosphate-5-kianse type Iγ (PIP5KIγ), which is recruited to focal adhesions by talin to locally generate PIP2 [66,67]. FAK is shown to interact with PIP2 [25] and is therefore very likely attached to the membrane in focal adhesions. Paxillin acts as an adaptor between the signalling and force transduction layer, by interacting with FAK and Src in the signalling layer and with vinculin in the force transduction layer [68,69,70]. The two main components of the force transduction layer are vinculin and talin. Talin is via its N-terminal FERM domain attached to cytoplasmic tails of β-integrin receptors and the membrane [40,71,72]. The long C-terminal rod domain of talin reaches across the force transduction layer and interacts via its most C-terminal helix with actin [73,74]. Vinculin acts as an enforcer of the linkage to actin by connecting the talin rod via several vinculin binding sites to actin [75,76].

### 3.2. FAK Attachement and Activation on the Membrane

FAK exhibits specificity for phosphoinositides that are phosphorylated on the D4 and D5 position of the inositol head group, while phosphorylation on the D3 position does not affect FAK affinity [26]. The basic KAKTLRK sequence in the FERM F2 lobe is required for PIP2 biding and it likely represents the primary PIP2 binding site in FAK. However, a second site on the kinase domain, that includes residues R508, K621 and K627, has recently been shown to also contribute to PIP2 binding [77]. In the autoinhibited conformation of FAK the two sites are positioned on perpendicular faces of the FAK surface, indicating that they would not be able to simultaneously form ideal interactions with a planer membrane bilayer. Coarse grain molecular dynamics simulations of autoinhibited FAK on a PIP2 membrane indicated that after membrane binding via the KAKTLRK sequence in the FERM domain the protein would rotate into two possible membrane bound states where FERM and kinase domains lie flat on the membrane with both domains bound to PIP2 [78]. However, in both states the KAKTLRK sequence is no longer fully engaged in PIP2 binding, while only in one of the predicted states the kinase interacts via the surface identified experimentally [77]. Large scale conformational changes that would allow for both experimentally identified PIP2 binding sites to interact simultaneously are likely beyond the scope of these simulations. Studies with a conformational FRET sensor have shown a significant conformational change to occur upon binding to PIP2 [26]. Such a conformational change is further supported by observations that autoinhibited FAK exhibits lower affinity for PIP2 compared to the Y180A, M183A mutant FAK where FERM and kinase domains are dissociated. A plausible interpretation is therefore that autoinhibited FAK undergoes an energy costly conformational change upon binding to a PIP2 membrane, which then allows interactions via both FERM and kinase domains with PIP2 membranes.

FAK mutated in the KAKTLRK sequence or FAK in PIP5KIγ knockdown cells still localise to focal adhesions, indicating that membrane interactions do not significantly contribute to FAK localisation [26,42]. On the other hand, KAKTLRK mutation and PIP5KIγ knockdown result in strong reduction of FAK activity, indicating that PIP2 binding is important for integrin mediated FAK activation. Experiments with purified FAK proteins show that PIP2 interactions do not directly increase catalytic turnover activity of FAK but trigger a multistep activation sequence resulting in active FAK [26]. Upon binding to PIP2 membranes FAK assembles into extended oligomers on the surface of PIP2 membranes, as shown by negative stain EM. The binding and oligomerisation then trigger conformational changes that expose the Y397 autophosphorylation site for efficient phosphorylation. It is not clear, however, whether the conformational changes remove autoinhibitory FERM-kinase interactions since the catalytic turnover activity is not increased upon PIP2 binding. An exact structural understanding of this membrane bound state of FAK and how it promotes autophosphorylation will have to await future high-resolution structural studies of FAK on PIP2 membranes. An intriguing question is how autophosphorylation can be efficient if turnover activity is not enhanced. A likely explanation provides the study of Grant and Adams [79] showing that, in kinase/substrate complexes and if the enzyme/substrate ratio is close to one, phosphorylation rates are not determined by steady-state kinetics as seen in regular turnover kinetics. Indeed, in autophosphorylation reactions every enzyme has to phosphorylate only one substrate, hence the off-rate of the product is irrelevant and no turnover is required. Likely, the oligomeric FAK assembly adopted on the membrane presents the autophosphorylation site of one FAK molecule to an active site of a neighbouring FAK molecule for efficient trans-autophosphorylation, even in absence of high turnover activity for exogenous substrates.

Regardless of the exact mechanism of PIP2 induced FAK autophosphorylation, as a consequence FAK recruits the Src kinase via SH2 binding to the phosphorylated Y397 site and SH3 binding to a proximal PxxP binding site on the FERM-kinase linker. In turn, Src phosphorylates Y576, Y577, Y861 and Y925 in FAK, the first two of which are located in the activation loop in the FAK kinase. Phosphorylation of the activation loop residues by Src is the key step inducing high turnover steady-state kinetics (see Section 2.2). An unresolved question today is whether the steps following autophosphorylation are further regulated or whether autophosphorylation will lead to FAK activation at high efficiency in all circumstances. Additional points of regulation could be Src recruitment and phosphorylation of FAK by Src. Clearly, Src phosphorylation of the activation loop is suboptimal in the autoinhibited state [35] and since conformational changes induced by membrane binding do not appear to expose the active site, it is conceivable that the Y576/Y577 phosphorylation sites adjacent to the active site are also not exposed. Structurally a plausible mechanism to unlock the catalytically inhibited state on the membrane and to expose the active site and activation loop residues, is stretching of the FAK molecule by force. Stretching forces are generated in focal adhesions by contracting actomyosin stress fibres attached to the actin regulatory layer. Such a mechanism could explain observations that have placed FAK as a prime candidate to play the role as a force sensor in focal adhesions, which translates mechanical force in focal adhesions into a biochemical signal (see Section 4). Firing of biochemical signals upon force generation allows the cell to respond in appropriate ways to different extracellular environments and ensures cell migration to occur in a coordinated manner.

## 4. FAK under Force

### 4.1. FAK as a Cellular Force Sensor

Contracting actomyosin stress fibres attached to the actin regulatory layer of the focal adhesion complex apply stretching forces to focal adhesion proteins anchored via multiple sites to the complex. The main components of the force transduction layer, talin and vinculin, represent important structural components in focal adhesions by connecting integrin receptors and the membrane proximal signalling layer to actin. It was demonstrated that these structural components experience average forces in the range of 2.5–10 pN [80,81] with peak forces reaching ~40 pN for molecules engaged with integrin receptors [82]. The stretching of talin is shown to reveal cryptic binding sites for vinculin, resulting in stabilization of the focal adhesion complex and a stronger attachment to actin [83,84]. Hence, the mechanobiology of structural focal adhesion components and how resulting conformational changes in the force transduction layer result in stabilisation and maturation of focal adhesions is relatively well characterised. In contrast, it is currently not understood how forces trigger biochemical signals in the signalling layer in focal adhesions.

FAK is a prime candidate to play the role as a signalling force sensor in focal adhesions. A number of cellular studies have demonstrated a relation between FAK activity and force generated in focal adhesions. Cellular strain, applied to cells by stretching of an elastic substrate the cells are growing on, directly increases the force experienced in focal adhesions and is shown to cause increased FAK phosphorylation levels in different cellular systems [28,29]. Further, plating of cells on substrates with different stiffness, where stiffer substrates allow the generation of higher forces in focal adhesions, shows that FAK activity increases on stiffer substrates [27]. The latter is also shown to occur in vivo, importantly in the context of tumour invasion where tissue stiffening is highly relevant in advanced tumours. Levental and colleagues [85] used a breast cancer mouse model to show that stiffening of the tumour stroma, induced by increased ECM crosslinking, results in activation of FAK which in turn causes a highly aggressive and invasive behaviour of tumour cells. In addition to be a responder to force, FAK also appears to provide sensory properties to cells. Normal cells when plated on a gradient of substrate stiffness can sense the stiffer regions and migrate towards them. This property is lost upon deletion of FAK [86]. Similarly, FAK is required for cells to respond to locally applied pulling forces that generate protrusions and focal adhesions at the pulling sites and reorient the cells towards them.

### 4.2. Structural Aspects of Force Mediated FAK Activation

Potential anchor points for the generation of stretching forces in the FAK molecule are at one end the FERM attachment to PIP2 membranes and at the C-terminal end the FAT interaction to paxillin, the connector to the structural components vinculin and talin. Via these attachments, contracting actomyosin fibres attached to talin and vinculin can be expected to transmit stretching forces to the FAK molecule. In agreement with such a scenario, vinculin is shown to move away from the membrane towards the actin regulatory layer during myosin II mediated focal adhesion maturation [87]. Further, using atomic force microscopy (AFM) based single molecule force spectroscopy, it was shown that forces applied to FAK molecules via the KAKTLRK lipid binding site in the FERM domain and the C-terminus of the kinase domain (the FAT domain was not included in these studies) results in rupture of autoinhibitory interactions between the FERM and kinase domain [88]. A force peak observed only with wild-type FAK but not the constitutively open mutant of FAK was observed at 25 pN using a retraction speed of the AFM cantilever of 12,800 nm/s. Absolute rupture forces depend on the loading rate with rupture occurring at higher forces with increased pulling speeds. The loading rate in focal adhesions is unknown at early stages of force application, but in mature focal adhesions, a steady state appears to be established with relatively constant force applied to the complex. Constant forces observed at 2.5–10 pN (with peak forces at 40 pN, see above) are therefore likely sufficient to cause rupture of FERM-kinase interactions and induce activating conformational changes. On the other hand, unfolding of the ATP loaded kinase domain was observed only above 50 pN, hence even peak forces in focal adhesions do not result in FAK deactivation. In order to obtain an atomic view of FAK stretching, experimental pulling of FAK by AFM was correlated to corresponding force-probe molecular dynamics simulations performed via the same attachment points [88]. Overall simulations agreed well with experiments, with FERM-kinase detachment occurring prior to any FAK unfolding over a range of loading rates. The interaction between the Y397 linker region and the FERM domain is mechanically shielded by the FERM-kinase interaction, hence FERM-kinase rupture is observed prior to FERM-linker detachment. In a separate study, atomistic molecular dynamics simulations were performed, starting from autoinhibited FAK bound via the KAKTLRK site to an explicit PIP2 containing membrane bilayer [89]. These simulations also suggested membrane interactions to occur via both, the FERM and kinase domains prior to force activation (see Section 3). Subsequent force application via the C-terminus of FAK then led to detachment of the kinase domain from FERM and membrane interactions, the order of which depended on the pulling angle. However, independent on pulling angle or loading rate, FERM-membrane interactions were always mechanically more stable than FERM-kinase or kinase-membrane attachments. Considering cellular observations together with the structural environment and attachment of FAK in focal adhesions it appears likely that stretching forces experienced by FAK contribute to maintaining FAK in an active state. To what extent force is important in FAK activation might however depend on multiple parameters and cellular context. To this end, in mouse embryo fibroblasts it was observed that FAK is efficiently autophosphorylated in nascent focal adhesions independent of force application [52].

Integrating known structural features involved in FAK regulation with effects of membrane binding and potential force activation, the following sequential scenario can be envisaged to switch FAK from the inactive to the active state (Figure 2): (1) FAT mediated localisation of autoinhibited FAK into focal adhesions promotes formation of FAK dimers by an increased local FAK concentration as well as potential dimer stabilisation by FAT-FERM interactions [44] (step 1 in Figure 2). (2) FAK dimers localised to focal adhesions attach to the nearby PIP2 rich membrane via the KAKTLRK basic patch, thereby removing the FAT domain from the FERM-KAKTLRK site. Membrane attachment is possibly enhanced by the increased avidity of FAK dimers. Membrane binding induces conformational changes in FAK, as shown using a conformational FERT sensor of FAK [26]. These conformational changes, possibly induced by kinase interactions with the membrane [77], promote FAK oligomerisation and expose the autophosphorylation site for efficient phosphorylation, while retaining low catalytic turnover activity [26] (step 2 Figure 2). (3) Force applied via structural focal adhesion components to the FAT domain of FAK could then be responsible for removing the kinase from FERM and membrane interactions to expose the activate site and the proximal activation loop for efficient phosphorylation by Src, triggering full FAK activation (step 3 in Figure 2).

In addition to FERM-kinase rupture, there are additional features in the structure of FAK that are compatible with force induced structural changes. As mentioned in Section 2.1, the FAK FERM domain displays an additional 10 residues at the C-terminus where other FERM and PTB domains interact with peptide ligands (coloured orange in Figure 1b). Stretching of the FAK molecule could cause release of these C-terminal FERM residues to liberate the site for potential interactions, such as cytoplasmic receptor tails. Equally, linker residues observed to interact with the FERM F3 lobe (Figure 1b), could be detached upon FAK stretching. This would liberate the PxxP sequence in the FERM-kinase linker and could promote Src recruitment via its SH3 domain.

Another intriguing thought is how force might affect the structure and function of the FAT domain. As discussed in Section 2.3, two paxillin LD motives can interact with two different faces of the FAT domain. With paxillin acting as a connector between FAK and the structural focal adhesion components, this interaction is likely responsible for transmitting force to membrane docked FAK molecules. Once sufficient strain is built up across the FAT domain, the N-terminal FAT helix would likely first detach from the bundle [56,57] (see Section 2.3). As a consequence, the paxillin LD motif interacting with the helix 1-4 face of the FAT domain would also detach. To some extent this might promote dissociation of FAK from focal adhesions [58]. However, mutational dissection of the two binding sites suggests the remaining paxillin interaction with the helix 2-3 face of the FAT domain to be sufficient for maintaining the majority of FAK-paxillin interactions [53]. In this scenario, force could be a trigger to release the N-terminal FAT helix resulting in exposure of Y925 for phosphorylation by Src. Phosphorylated Y925 is a docking site for the adaptor protein Grb2 [90], which on the one hand links FAK signalling to the Ras/MAPK signalling pathway but was also suggested to recruit the large GTPase dynamin to FAK and contribute to integrin internalisation and disassembly of focal adhesions. In addition to direct FAT-paxillin detachment, this mechanism might contribute to the increased cytosolic FAK localisation seen with a mutant that destabilises the N-terminal FAT helix [58].

## 5. Conclusions and Future Challenges

High resolution structures of FAK regions, including all portions predicted to adopt globular domains, have provided important insights into the architecture and function of FAK at atomic level. These structures have defined the mechanism of FAK autoinhibition in the cytosol [35] and identified a number of interactions, including a mode of recruitment into focal adhesions via paxillin [53,54] and FERM mediated dimerization important for activation [44]. Important future challenges include a structural understanding of how FAK integrates in its various cellular environments and how this leads to structural rearrangements that affect FAK activity, its interaction network or localisation. Analysing FAK structure and interactions in the crowded and dynamic environment of the focal adhesion complex will be a major future challenge. Super resolution optical microscopy has pinned down the ultrastructural localisation of FAK in focal adhesions to a membrane associated signalling layer [24]. Two complementary approaches can be envisaged to provide in the future data that can bridge the resolution gap from the nanoscale to an atomistic description: (1) Structural studies on reconstituted focal adhesion complexes assembled on membrane model systems using purified components and (2) In situ cellular structural analysis of focal adhesions using cryo-electron tomography. Major challenges of the former are the membrane environment, the complexity and the high dynamics during focal adhesion maturation. Regarding membrane systems, nanodiscs have proven a suitable system for high resolution cryo-EM studies [91]. However, current forms used might not provide sufficient membrane surface for the assembly of extended focal adhesion complexes. The force applied on focal adhesion complexes induces highly dynamic conformational changes, hence atomic descriptions might require combining structural techniques with force spectroscopy and computational modelling. Reconstitution studies will likely require a reductionist approach, focusing on the core focal adhesion components, rather than the full adhesome of the order of ~150 proteins [20,21]. To this end, a consensus adhesome was defined by comparison of various proteomics studies [19].

On the other hand, studying the FAK structure in the context of the focal adhesion ultrastructure in cells by cryo-electron tomography will also entail major challenges. A past study has identified donut shaped structures in focal adhesions [92]. However, due to the crowded environment a major difficulty is the assignment of recognised shapes to specific focal adhesion proteins. Reaching sub-nanometer resolution could greatly facilitate this process, however this would likely require extensive sub-tomogram averaging of repeated structural features. Due to the challenges described above, progress will likely come stepwise in the form of substructures and will perhaps lag behind the explosion of high resolution cellular mega-complexes being reported in the post genomic area as a result of the recent revolutionary advances in cryo-electron microscopy [93]. Being able to visualise the FAK protein in its focal adhesion environment at different stages of the life cycle in focal adhesions could greatly help to answer several of the currently rather speculative models discussed in this review.

## Figures and Tables

**Figure 1 biomolecules-10-00179-f001:**
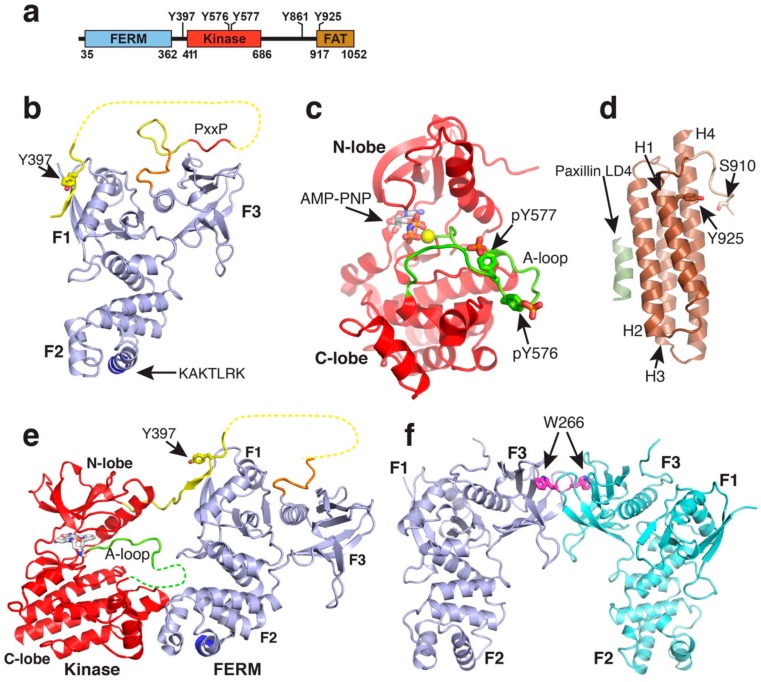
Structures of FAK domains. (**a**) Schematic domain structure of FAK. Domain boundaries and regulatory phosphorylation sites are indicated. (**b**) Crystal structure of the FAK FERM domain containing the F1, F1 and F3 lobes (PDB accession 2AL6). The 10 C-terminal residues integral to the FAK FERM domain, but not in ERM FERM domains, are coloured in orange. The linker is in yellow with the PxxP motif known to interact with the Src SH3 domain in red. The Y397 autophosphorylation site in the linker and the KAKTLRK motif in the F2 lobe (blue) are labelled. (**c**) Structure of the active FAK kinase domain with Y576 and Y577 in the activation loop (A-loop) phosphorylated (PDB accession 2J0L). The A-loop is coloured in green. A non-hydrolysable ATP analogue (AMP-PNP) and a Mg^2+^ ion (yellow sphere) are bound to the active site. (**d**) Structure of the FAT domain bound to the paxillin LD4 peptide (PDB accession 1OW7). The paxillin LD4 peptide is in olive green and the N-terminal FAT extension in tan. Helices H1-H4 and the S910 and Y925 phosphorylation sites are labelled. (**e**) Crystal structure of the FERM-kinase region of FAK in the autoinhibited conformation (PDB accession 2J0J). Colouring is as in panels (**a**,**b**). (**f**) FERM dimer mediated by W266 interactions in the F3 lobe as observed in various crystal structures (shown from PDB accession 2AEH).

**Figure 2 biomolecules-10-00179-f002:**
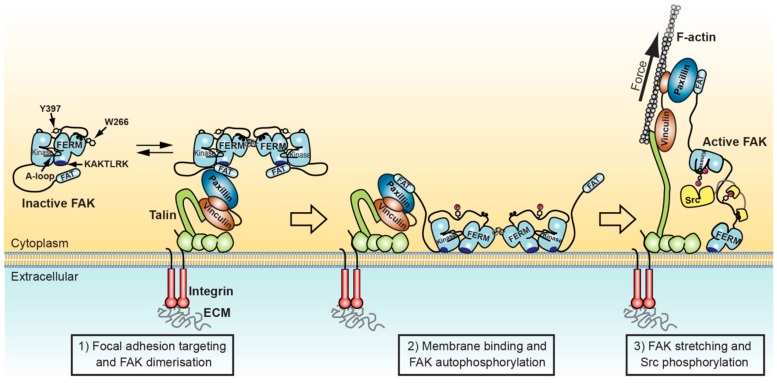
Model for FAK activation in focal adhesions mediated by membrane interactions and force. Left: FAK can form dimers via F3 lobe interactions involving W266. FAK dimer formation is likely promoted upon recruitment into focal adhesions by increased local concentration and potentially by paxillin-FAT interactions that synergise with FAT-FERM interactions to stabilise FAK dimers. Middle: FAK interacts with nearby PIP2 rich membranes via the KAKTLRK basic patch in the FERM F2 lobe. Increased avidity for PIP2 of FAK dimers might enhance membrane attachment. Membrane interactions induce conformational changes that allow simultaneous interactions of FERM and kinase domains to the membrane and expose the FERM-kinase linker for efficient autophosphorylation on Y397. Right: Src binds with its SH2 domain to the phosphorylated Y397 site and with the SH3 domain to the PxxP motif in the FERM-kinase linker. Stretching forces applied to FAK by contracting actomyosin fibres cause release of the FAK kinase domain from FERM and membrane interactions, exposing the activation loop for efficient phosphorylation by Src, resulting in FAK activation.

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
