# Peer review of "FAK Structure and Regulation by Membrane Interactions and Force in Focal Adhesions"

_biomolecules, 2020, doi:10.3390/biom10020179_

Round 1

Reviewer 1 Report

This review from Daniel Lietha’s group pieces together a detailed model about FAK domain structure and conformational activation mechanisms. Focus is on the conformational changes induced by phospholipid binding and via indirect protein interactions that connect FAK to the actin cytoskeleton.

This is a timely review as Lietha’s group has recently published some strong supportive papers in this area. 

The review could be strengthened if the author’s could provide consideration to the points listed below:

A) General: please cite some current reviews that cover the roles of FAK outside of focal adhesions (endosomes, nuclei, adherens junctions, etc).

1) Section-4, FAK under Force

The authors group force transduction at focal adhesions triggering FAK activation from studies that are investigating both outside-in and inside-out integrin activation events. Please consider discussing these linkages separately.

Some recent papers [Chang et al. PNAS 116:3524, 2019 regarding FAK-RIAM}

and add data with regard to emerging role for kindlin in FAK activation [Theodosiou et al. eLife 2015, 5:e101030]

2) It is also requested that the authors provide some additional discussion of studies that do not necessarily fit within their model.

Papusheva et al., J. Cell Science 122:656, 2008 - that FAK biosensors are conformational sensors and not necessarily connected to changes in FAK activity.

Lawson et al., J. Cell Biol. 196: 2:223, 2012 - that increased FAK Y397 can occur in early adhesions in a tension-independent manner and that mutation of talin binding to FAK does not disrupt fibronectin adhesion-mediated FAK activation (Y397 phosphorylation).

3) References. The papers chosen by the authors are a mix of original findings from the early 1990’s to some recent studies that have not necessarily withstood the test of time.  It is suggested that the authors reconsider some tangentially-linked references throughout.

Some specific reference inconsistencies -

#55. It was shown by Wu et al, Oncogene 27: 1439, 2008 that FAK directly phosphorylates Src at Tyr-416 within the catalytic domain enhancing Src activation.

#72. The paper cited is for protein kinase A.  A better reference would be Schneck et al. Biochemistry 49: 7151, 2010 “Kinetic mechanisms and rate limiting steps of focal adhesion kinase-1”

line 55. Cross-talk “that” also feed into…

line 246 and 364. FRET (not FERT)

Figure 1d. The graphic positions S910 near Y925.  It is not clear that this occurs. Suggest remove S910 labeleing.

line 272. Need to re-write sentence to convey that FAK Y397 phosphorylation can occur in trans within a FAK multimer.

Figure 2.  Suggest alterations that highlight Force independent (early) and Force-dependent steps in the FAK activation model.  As shown, it is inferred that FAK activation does not occur until force transduction (3).  Cell studies show that FAK Y397 phosphorylation can occur early upon initial integrin clusterin (1) that is mostly force independent.

Author Response

Response to reviewer 1:

We thank the reviewer for the interesting comments and suggestions. Please find below our point-by point responses.

“A) General: please cite some current reviews that cover the roles of FAK outside of focal adhesions (endosomes, nuclei, adherens junctions, etc). “

We now point in the introduction to the role of FAK outside focal adhesions (lines 61-62) and cite the following review that discusses the role of FAK in endosomes, nucleus and adherens junctions: Kleinschmidt EG, Schlaepfer DD. Focal adhesion kinase signaling in unexpected places. Curr Opin Cell Biol. 2017. We also clarify that in our review we specifically focus on structural changes of FAK upon integration into focal adhesions (line 69).

“1) Section-4, FAK under Force

The authors group force transduction at focal adhesions triggering FAK activation from studies that are investigating both outside-in and inside-out integrin activation events. Please consider discussing these linkages separately.”

 We thank the reviewer for this suggestion, it is an important point. We added a paragraph explaining the two different mechanisms and add related references (lines 46-50).

With respect to FAK activation by membrane interactions and force we think this separation could add an additional layer of complexity that might confuse a lay reader. Although the initial upstream path leading to FAK activation is likely different depending on whether integrins are activated by an outside-in or inside-out mechanism, it is currently not known whether the final structural trigger resulting in FAK activation is different. Both mechanisms place FAK in close proximity to the membrane, hence in both PIP2 mediated effects could occur. Further, the effect of force likely only plays a role at a stage where the full ECM-integrin-actin linkage is established, at which point the two mechanisms converge.

“2) It is also requested that the authors provide some additional discussion of studies that do not necessarily fit within their model.”

We thank the reviewer for this suggestion. We added discussions involving the two mentioned studies (lines 228-232 and lines 378-381)

“3) References. The papers chosen by the authors are a mix of original findings from the early 1990’s to some recent studies that have not necessarily withstood the test of time. It is suggested that the authors reconsider some tangentially-linked references throughout.”

 We added several additional references throughout the manuscript.

“Some specific reference inconsistencies -

#55. It was shown by Wu et al, Oncogene 27: 1439, 2008 that FAK directly phosphorylates Src at Tyr-416 within the catalytic domain enhancing Src activation.”

We use reference #55 (now #60) to refer to the finding that Src can be recruited into FAs by FAK.

“#72. The paper cited is for protein kinase A. A better reference would be Schneck et al. Biochemistry 49: 7151, 2010 “Kinetic mechanisms and rate limiting steps of focal adhesion kinase-1””

Schneck et al. present steady-state kinetics. We would like to specifically point out that autophosphorylation does not follow steady-state kinetics. To our knowledge this is best demonstrated for PKA.

“line 55. Cross-talk “that” also feed into... “

This is now corrected.

“line 246 and 364. FRET (not FERT)”

This is now corrected.

“Figure 1d. The graphic positions S910 near Y925. It is not clear that this occurs. Suggest remove S910 labeleing.”

We added the notion “It is not clear whether this interaction occurs natively …” (line 152), but prefer to keep the labelling since we refer to the residue in the text and a significance is described for this phosphorylation site.

“line 272. Need to re-write sentence to convey that FAK Y397 phosphorylation can occur in trans within a FAK multimer.”

This is now corrected.

“Figure 2. Suggest alterations that highlight Force independent (early) and Force-dependent steps in the FAK activation model. As shown, it is inferred that FAK activation does not occur until force transduction (3). Cell studies show that FAK Y397 phosphorylation can occur early upon initial integrin clusterin (1) that is mostly force independent.”

We added a paragraph mentioning that Y397 phosphorylation is seen in early adhesion structures independent of force (lines 378-381). We also mention that the role and relevance of force might be context specific. In section 3 (page 7, bottom paragarph) we further present a force independent model where autophosphorylation leads to Src recruitment and phosphorylation by Src. In this paragraph we propose that additional levels of control could exist such as force. According to our model force would likely promote Src phosphorylation of Y576/Y577, whereas Y397 phosphorylation (which is not catalytically activating) is promoted by membrane interactions and is independent of force.

Reviewer 2 Report

This is an excellent well-written review article authored by Martinez et al. titled "FAK structure and regulation by membrane interactions and force in focal adhesions." A signficant overview of multiple X-ray crystal structures of FAK is representated as well as the multiple mechanisms which regulate FAK activation. The figures are well-presented with high-quality illustrations. A few minor issues should be addressed to complete this compelling manuscript:

1) Expansion on the interplay between the FAT and FERM domains would be useful in the explanation of the FAK activation model via force. Specifically, it is unclear if the authors depict disruption of FERM-FAT binding after FERM-PIP2 interactions, therefore allowing FAT to be "stretched" by contracting actin fibers. Also, since FAT is known to be required for FA localization, what is the distinction between FAT-mediated localization vs. FERM-mediated localization?

2) Absent from the manuscript is the discussion of a new FAK FERM crystal structure (PDB 6CB0) which identified an alternative conformation of the PxxP motif bound to a cleft between the F1 and F3 lobes.

3) It would be more specific to name the two LD-motif binding sites on the FAT domain (Helix 1-4 vs Helix 2-3).

4) There is no discussion regarding the direct phosphorylation of FAK at Y397 by growth factor receptors.

Author Response

Response to reviewer 2:

We thank the reviewer for the interesting suggestions and comments. Please find below our point-by point responses.

“ 1) Expansion on the interplay between the FAT and FERM domains would be useful in the explanation of the FAK activation model via force. Specifically, it is unclear if the authors depict disruption of FERM-FAT binding after FERM-PIP2 interactions, therefore allowing FAT to be "stretched" by contracting actin fibers. Also, since FAT is known to be required for FA localization, what is the distinction between FAT-mediated localization vs. FERM-mediated localization?

Indeed, since both FAT and PIP2 interact with the same KAKTLRK sequence we think that membrane binding will displace the FAT-FERM interaction. We mention this in lines 223-224 and also added an explicit explanation of this where we describe the model presented in Fig.2 (lines 387-388).  

As for FA localization we think that FAT is predominantly responsible for FA localisation. We mention in lines 143-144 that FAT is responsible for FA targeting and in lines 272-273 that FERM-membrane interactions are not sufficient for FA localisation (supported by references 26 and 42).

“2) Absent from the manuscript is the discussion of a new FAK FERM crystal structure (PDB 6CB0) which identified an alternative conformation of the PxxP motif bound to a cleft between the F1 and F3 lobes. “

We thank the reviewer for pointing this out. In fact one FERM molecule in 6CB0 makes the same PxxP interaction with the F3 lobe as seen in 2AL6 and in the other the PxxP binds to the cleft. We explain this now in lines 93-95.

“3) It would be more specific to name the two LD-motif binding sites on the FAT domain (Helix 1-4 vs Helix 2-3).”

We thank the reviewer for this suggestion, we changed the nomenclature accordingly.

 “4) There is no discussion regarding the direct phosphorylation of FAK at Y397 by growth factor receptors.”

We mention this now in lines 216-218.

Reviewer 3 Report

The authors make a comprehensive review about FAK. This manuscrpit discuss regulatory mechanisms regarding FAK activation too.  In general, is a well written review, well structured and providing concise infromation.

The only minor change that should be corrected is that the manuscript need slight english review

Author Response

Response to reviewer 3:

We thank the reviewer for the positive comments. We have given the manuscript to a native English person for review and introduced several corrections throughout.